# A Data-Driven Factor Graph Model for Anchor-Based Positioning

**DOI:** 10.3390/s23125660

**Published:** 2023-06-17

**Authors:** Ana Moragrega, Carles Fernández-Prades

**Affiliations:** Centre Tecnològic de Telecomunicacions de Catalunya (CTTC/CERCA), 08860 Castelldefels, Barcelona, Spain

**Keywords:** anchor-based positioning, belief propagation, factor graph, lateration, weighted geometric dilution of precision, weighted least squares

## Abstract

This work presents a data-driven factor graph (FG) model designed to perform anchor-based positioning. The system makes use of the FG to compute the target position, given the distance measurements to the anchor node that know its own position.The aim was to design a hybrid structure (that involves data and modeling approaches) to address positioning models from a Bayesian point of view, customizing them for each technology and scenario. The weighted geometric dilution of precision (WGDOP) metric, which measures the effect on the positioning solution of distance error to the corresponding anchor node and network geometry of the anchor nodes, was taken into account. The presented algorithms were tested with simulated data and also with real-life data collected from IEEE 802.15.4-compliant sensor network nodes with a physical layer based on ultra-wide band (UWB) technology, in scenarios with one target node, three and four anchor nodes, and a time-of-arrival-based range technique. The results showed that the presented algorithm based on the FG technique provided better positioning results than the least squares-based algorithms and even UWB-based commercial systems in various scenarios, with different setups in terms of geometries and propagation conditions.

## 1. Introduction

Reliable and accurate positioning is mandatory for location-based services such as intelligent transportation systems, self-driving cars, unmanned aerial vehicles, the Internet of Things, and indoor applications [1]; as well as for enhanced communications and mobility management in 5G and beyond [2]. Satellite-based navigation systems can be considered the technology of choice in outdoor environments for positioning and navigation. However, in dense urban areas or urban canyons with the presence of strong multipath echoes or even total blockage of global navigation satellite system (GNSS) signals, position accuracy is degraded or, in indoor scenarios or tunnels, not available. The data fusion of hybrid approaches with multiple sensors and technologies can address many of the vulnerabilities of satellite-based systems in these environments [3,4]. Therefore, different combinations of sensors and technologies are suitable for positioning and its applications, depending on the environment, dynamics, budget, accuracy, requirements, and degree of robustness or integrity required [1].

(1)State of the art and motivation:

Regarding data fusion for positioning, there are different approaches to integrating data, which face technical challenges for optimal weighting [1]. Some challenges may be due to the characteristics of the technology used (e.g., propagation conditions for radio based technologies), which can affect the results.

In the case of challenging scenarios, such as complex indoor setups, multipath and non-line-of-sight (NLOS) effects are known to cause errors in ranging and positioning. There have been different approaches in the literature of recent years to overcome this problem; however, it is a subject that remains open. For example, some works focused on the development of localization systems hybridizing data from different sensors or technologies [3,5]. Other works focused on the identification of NLOS or multipath errors. As an example, machine learning (ML) methods for NLOS identification and mitigation in localization with ultra-wide band (UWB) experimental data were applied [6]. ML and deep learning methods have been explored and applied to ranging and positioning, to face some of these challenging scenarios [7]. In recent years, algorithms based on factor graphs (FG) have also been proposed to improve positioning and navigation. As an example, an overview of the state of the art in recent years of FG-based navigation and localization was presented in [8], while a FG for vehicle cooperative localization was proposed in [9]. The algorithm introduced a FG-based solution that added the topology (inter-vehicle distance) as a constraint. In the robotics field, FG optimization was presented in [10]. A FG was proposed for GNSS/INS integration in [11]. These works were based on previous articles that presented FG-based modeling: in [12], FG based techniques for modeling signal processing algorithms were explained, and a message passing algorithm (belief propagation or sum-product algorithms) for FG model resolution was presented. In [13], the author explained how inference algorithms (message passing) with graphical models such as Bayesian networks (BN) and FG are a powerful tool to model scenarios of interest (this methodology was called model-based machine learning). Using this FG modeling, in this present work, we propose a data-driven FG model whose purpose is to perform anchor-based positioning.

Anchor-based positioning may involve lateration methods; that is, the use of distances or ranges for determining the unknown position coordinates of a node [1]. In this case, in anchor-based positioning, the algorithms estimate the target (tag) node position (positioning phase) given the distance measurements (raging phase) to the anchor nodes that know its position. In general, works in the literature (regarding anchor-based positioning with lateration) have addressed challenging conditions in the propagation between the target and anchor nodes in the ranging phase. In [7], an overview of the state of the art of the recent years in machine learning for indoor positioning was presented. In [14], an NLOS mitigation algorithm for UWB localization in harsh indoor environments was proposed. In most of these cases, the solutions focused on identification/mitigation of challenging propagation conditions in ranging. Thus, these previous solutions did not take into account both the distance error (due to propagation errors) and the geometry effects (due to the geometrical situation of the anchor nodes) in the ranging and positioning phases. However, there are some solutions that address this challenge. As examples, there are techniques such as the weighted least squares (WLS) algorithm and other algorithms that address this issue in commercial nodes. Regarding WLS-based algorithms, they obtain a solution that allows assigning a weight to the distances inversely proportional to their error covariance, while providing a mathematical interpretation of the effect of the anchor nodes geometry on positional measurement precision. With respect to commercial UWB nodes (DW), modules such as the DWM1001 (MDEK1001 system [15] from Decawave, Qorvo) provide a solution to this challenge.Their solution is based on a previous step that selects anchor nodes managing NLOS conditions during transmission and the geometric issue before a maximum likelihood positioning algorithm. However, the WLS and DW approaches are not sufficient for scenarios with challenging propagation and geometric conditions. Moreover, they do not provide the advantages of FG related to Bayesian inference, covariance estimation, and so on.

In this work, a factor graph model for anchor-based positioning is proposed that takes into account both the effect of the propagation conditions and the effect of geometry involved in this type of positioning. As far as we know, there have not been any FG models that considered these two effects for anchor-based positioning and navigation.

(2)Why data-driven factor graph models?

This work presents algorithms designed with Bayesian inference [13], implemented through probabilistic graphical models and using message passing algorithms [16]. Therefore, graphical models are considered, such as Bayesian networks [17] handled as FG with a belief propagation (BP) (or sum-product) algorithm [18]. A FG is a bipartite graph that allows the graphical representation of the dependencies between variables and factors. A brief introduction to FG is presented in Appendix A. Using this framework, a data-driven factor graph model was designed, to perform anchor-based positioning.

The propagation conditions (e.g., multipath or LOS/NLOS) and geometry conditions (i.e., the geometry that forms the placement of the anchor nodes) that affect ranging and positioning errors are scenario dependent. Therefore, the aim was to design a hybrid structure (that involves scenario data and model approaches) to address positioning models from a statistical Bayesian point of view, customizing them for each technology and scenario. Moreover, the design requires flexibility to adequately adapt the algorithm to the technologies used in each case. Thus, a data-driven FG algorithm provides this hybrid and flexible modeling. Therefore, the FG model for positioning learns from the input data, which are different for each scenario. This is a data-driven FG model where an algorithm iterates, converging on the optimal solution. The convergence of the BP is ensured, since the FG has been modeled as a tree, a connected graph without cycles. BP and Gaussian BP converge in singly connected graphs but may fail to converge in graphs with loops [19]. Moreover, the FG is a linear model with Gaussian BP (Gaussian aleatory variables), which simplifies the calculation of the integrals of the messages between the factors and variables that form the FG (Appendix A). For Gaussian BP with linear relations between variables, the messages of the FG can be represented by the mean vector and the covariance matrix of the corresponding Gaussian random variable [16].

Regarding wireless radio technologies, there are different ranging methods, mainly based on received signal strength, time of arrival (TOA), and angle of arrival [1,20]. The main advantages of UWB technology with TOA-based ranging techniques are the high accuracy, the low complexity, and the low budget compared to other wireless radio technologies. Typically, the accuracy is less than 10 cm in a good scenario for positioning purposes. In the presented case, distance estimation is performed with radio devices with ultra-wide band (UWB) technology (IEEE 802.15.4 standard [21] with a physical layer based on UWB), where transmitted signals are used to estimate the range. Hence, the anchor-based positioning is based on an integration architecture, in which ranging measurements from UWB anchor nodes (that know its position) to the UWB target (tag) node are combined using a designed FG to estimate the position of the target node. Weighting the ranges requires the standard deviation (or covariance) error. Covariance values are provided by statistics of UWB measurements in case of collected data being provided. Linear systems and Gaussian distributions are assumed for the FG. Thus, the ranging model is linearized in advance. The design of this FG-based algorithm has been customized for IEEE 802.15.4 technology with a physical layer (PHY) based on UWB.

In this work, a FG model was designed avoiding loops. The algorithm was introduced in [22]; however, in this present work, the algorithm is explained with new contributions detailed in Section 1.2. The proposed algorithm has a first stage in which the distances to the anchor nodes (i.e., random variables) are grouped; on the one hand, to avoid loops in the FG and, on the other hand, so that the position solution of each group is weighted on the basis of its covariance. In this part, the estimated covariance is related to the weighted geometric dilution of precision (WGDOP) metric, which measures *(i)* the effect on the positioning solution of the distance error from the tag to the anchor nodes and *(ii)* the geometry formed by the position coordinates of the anchor nodes with respect to the target node. As a result, the target position is estimated using weighting techniques, in which the position solution of each group is weighted on the basis of its covariance related to both transmission and geometry conditions. Moreover, the weighting approach allows the distances with the best covariance results to contribute more than the others to position estimation.

The results show that the proposed algorithm based on the FG technique provides better positioning results than the least squares (LS) and WLS algorithms and commercial DW UWB nodes in different scenarios with different geometries and propagation conditions from the anchor nodes. This algorithm and framework could be applied to systems based on anchor-based positioning with different technologies and with different ranging and positioning techniques, as well as in different scenarios, such as for urban or indoor positioning and with mobile targets such as unmanned aerial vehicles, and so on. For these cases, the FG could be changed and adapted, as it is a flexible tool.

### 1.1. Main Objectives

The design, development, and testing of a hybrid structure (which involves data and modeling approaches) to address models for positioning from a Bayesian point of view, customizing them for each technology and scenario;The study and development of techniques based on machine deep learning and specifically factor graph modeling to improve positioning in good, intermediate, and challenging scenarios. In the presented case, anchor-based positioning techniques with lateration using radio devices (UWB, IEEE 802.15.4) were considered;To use both simulated data and collected real data to test the data-driven factor graph model and its convergence. The algorithm was tested with simulated data in [22]; however, the present work evaluated and studied the algorithm more completely with simulations. Additionally, real-life data from a collected and published dataset [23] were used to obtain more complete results. The dataset was based on data collected from commercial UWB-based devices, in benign, intermediate and, challenging positioning scenarios.

### 1.2. Main Contributions and Outline

Section 2 describes the proposed system model for anchor-based positioning and for the FG: nodes, scenarios, WGDOP metric, ranging model, and positioning with least squares methods. The considered FG is a linear system; thus, the ranging model is linearized with the Taylor series and an iterative method is introduced in the FG;Section 3 presents the proposed FG algorithm, which avoids loops. This work is a more complete extension of the paper in [22]. Therefore, there are parts of this work similar to the prior paper. However, there are many new contributions. Thus, for convenience the algorithm is presented as in [22]. However, in this current work, the FG is explained in more detail with the messages of BP (Section 3.3), the pseudocode, and details of the iterative algorithm. Moreover, in that section, the grouping of distances is explained and also how the position solution of each group is weighted based on its covariance related to the WGDOP metric. The iterative method is explained in the pseudocode of the algorithm (Section 3.4). The FG-based algorithm is a data-model hybrid structure, whose model learns from the data. It is an iterative algorithm, until converging on the solution;Section 4 details the results with real and simulated data and the convergence of the algorithm to an optimal solution for both cases. Although the algorithm was presented in [22], it was only tested with simulated input data. The present work evaluated and studied the algorithm more completely with simulations. Additionally, real-life data from a collected and published dataset [23] are used to obtain results. The dataset is based on data collected from commercial UWB-based devices, in benign, intermediate, and challenging positioning scenarios. The positioning results of the algorithm show that the presented FG algorithm achieved better results than an UWB commercial solution and classical approaches (iterative LS and WLS algorithms) in various scenarios with different conditions in terms of geometry and propagation conditions for anchor-based positioning.

In [22], weighting values of the FG for each distance were estimated using simulated data. In the present work, the weighting values were estimated with both simulated data and real collected data. Furthermore, the effect on the position estimation of the grouping of variables was evaluated with the WGDOP metric and with collected data.

Section 6 draws some conclusions.

## 2. System Model for Anchor-Based Positioning

### 2.1. Scenarios

The scenarios of interest involve anchor-based positioning, in which there are two types of node: anchor nodes that know its position and a target node that does not. One is interested in knowing the probability of the target position x given the measurements $m_{i}$ of distance from the anchor nodes to the target node. Thus, the posterior probability density function, which can be approximated as $p\left(x|m_{1},\ldots ,m_{N}\right)\propto p\left(x\right)\prod_{i=1}^{N}p\left(m_{i}|x\right)$, is used. The two-dimensional coordinates (M=2) of the nodes are defined as $x={\left[x,y\right]}^{T}$ for the target node, and $x_{a_{\left(i\right)}}={\left[x_{a_{\left(i\right)}},y_{a_{\left(i\right)}}\right]}^{T},i=1,\cdots ,N$ for the anchor nodes.

In this data fusion model for cooperative and hybrid positioning, the nodes are able to estimate ranges using different technologies. However, for this work, wireless technology based on the IEEE 802.15.4 standard [21] was considered with a physical layer based on UWB, and ranging techniques based on time of arrival. The geometric distance between the target node and the *i*-th anchor is defined as $\varrho_{i}\left(x\right)=\left\|x-x_{a_{\left(i\right)}}\right\|$, where $\varrho_{i}\left(x\right)$ is estimated with a ranging protocol. The IEEE 802.15.4-2011 [21] version with a UWB-based PHY layer (and previous versions) defines a mandatory ranging protocol called two-way time-of-arrival (TW-TOA) and an optional symmetric double-sided (SDS) TW-TOA protocol that reduces the effect of an imperfect timing reference. The medium access control layer (MAC) is based on a superframe structure, in which nodes have a reserved time slot for the sending and reception of frames. These frames are intended to perform the ranging protocol, as well as data communication and standard primitives.

The presented algorithms were tested with simulated and real-life data with UWB devices. Therefore, data collected from the published dataset in [23] were used. Data were collected from IEEE 802.15.4-2011 [21] compliant nodes with a PHY layer based on UWB technology. Nodes were mounted in good, intermediate, and challenging scenarios for positioning purposes. Anchor-based positioning can be affected by propagation effects (e.g., bias or multipath propagation) from the anchor to the tag node (in the ranging protocol), as well as the geometry formed by the anchor nodes that estimate the distance from the tag. Therefore, the WGDOP parameter was used to study how conditions related to the distance estimation error and the geometry effects that form the anchor nodes affect the position estimated by the tag node. In [24] and [20] (pp. 33–35), the WGDOP parameter is explained in more detail. WGDOP is the GDOP parameter accounting for the quality of each range measurement.

The WGDOP metric is explained in [20]. WGDOP involves an inverse visibility matrix H (defined in Section 2.2) relating the covariance of range errors to that of a position solution. Furthermore, a conceptual representation of the concept behind WGDOP is shown in Figure 1, where two anchor nodes are used to solve for a two-dimensional position in the plane, using range measurements with lateration. In the presence of noisy ranges, the uncertainty is visualized as two concentric circles, with the true range lying in between. The intersection of the two circles, in the noisy case, provides an area in which the receiver is estimated to be. When comparing Figure 1a,b, it becomes clear that the geometrical situation of the anchor nodes affects the size of this area, which is indeed quantified by the WGDOP.

### 2.2. Linearized Ranging Model

In this work, the geometric distance $\varrho_{i}$ between nodes is estimated with a TOA-based technique considered in the standard IEEE 802.15.4 with UWB technology. Measurements $m_{i}={\underline{d}}_{i}$ are modeled with the distance $\rho_{i}$ between the target and the anchor node *i* following this equation:(1)$$\rho_{i}=\varrho_{i}\left(x,x_{a_{\left(i\right)}}\right)+\upsilon \;,$$
where $\upsilon ∼N(0,\sigma_{{\underline{d}}_{i}}^{2})$. In the present case, the ranging model does not consider the bias error. The errors in distance estimation with TOA-based techniques may have different sources. For example, the signal traveling between nodes can be affected by the environment and obstacles, e.g., objects, people, walls, or cars.

The FG is modeled as a linear system with range models that relate positions to estimates ${\underline{d}}_{i}$. However, the models in (Equation 1) are not linear. Thus, the previous equation is linearized using a Taylor series, so that (Equation 1) may be
(2)$$\rho_{i}=\rho_{i}^{0}+\left(\frac{x^{0}-x_{a_{i}}}{d_{i}^{0}}\right)\left(x-x^{0}\right)+\left(\frac{y^{0}-y_{a_{i}}}{d_{i}^{0}}\right)\left(y-y^{0}\right)\;,$$
where ${(}^{0})$ indicates initial values for each iteration of the algorithm. Once (Equation 1) is linearized and the terms are rearranged in (Equation 2), one obtains the linear model (Equation 3) that is used to relate certain random variables (modeled with Gaussian distributions) of the factor graph.

### 2.3. Positioning with Least Squares Methods in a Linear Setting

From (Equation 2), one can obtain the following known linear model [24] for TOA-based ranging with *N* anchor nodes and anchor-based positioning
(3)Δd=H·Δx,
where $\Delta d={\left[\Delta d_{1},\ldots ,\Delta d_{N}\right]}^{T}$, $\Delta x={\left[\Delta x,\Delta y\right]}^{T}$, $\Delta d_{i}={\underline{d}}_{i}-d_{i}^{0}$, $\Delta x=x-x^{0}$, $\Delta y=y-y^{0}$ and where the elements of H are $a_{\left(i=1\ldots N,m=1\right)}=\frac{x^{0}-x_{a_{i}}}{d_{i}^{0}}$ and $a_{\left(i=1\ldots N,m=2\right)}=\frac{y^{0}-y_{a_{i}}}{d_{i}^{0}}$. In order to obtain the position of the target node, the system (Equation 3) can be solved applying a pseudo inverse as in the least squares problem if N≥M as
(4)$$\Delta x={\left(H^{T}H\right)}^{-1}H^{T}\Delta d\;.$$

The system can be solved iteratively both for the LS and WLS algorithms. The latter involves solving a WLS problem in each step:(5)$$\Delta x={\left(H^{T}WH\right)}^{-1}H^{T}W\Delta d\;,$$
where W is the diagonal matrix of weights $w_{i}=\sigma_{{\underline{d}}_{i}}^{-2}$.

The covariance of the position with WLS estimation $C_{\hat{x}}={\left(H^{T}WH\right)}^{-1}$ is related to the WGDOP as $\mathrm{WGDOP}=\sqrt{Trace{\left(H^{T}WH\right)}^{-1})}\;,$ which is the WGDOP of the UWB tag node related to the network of reference UWB nodes in this range [20].

## 3. Factor Graph Model for Anchor-Based Positioning

The variables of the previous linearized ranging and positioning algorithms (Section 2) were used and related through the Bayesian network (BN) of Equation (Equation 6). A BN is a probabilistic graphical model that represents a set of variables and their conditional dependencies via a directed acyclic graph. Therefore, the probability of the position x of the target given the distance measurements is the following BN:(6)$$p\left(x|{\underline{d}}_{1},\ldots ,{\underline{d}}_{N}\right)\propto p\left(x\right)p\left(y\right)\prod_{i=1}^{N}p\left({\underline{d}}_{i}|\Delta d_{i}\right)\cdot p\left(\Delta d_{i}|\Delta x\right)p\left(\Delta x|x\right)p\left(\Delta d_{i}|\Delta y\right)p\left(\Delta y|y\right).$$

Appendix A provides an introduction to BN and FG, detailing the main steps from BN to FG solving with a message passing algorithm (belief propagation, also known as a sum-product algorithm).

Figure 2 and Figure 3 show the set of variables of (Equation 6) and their dependencies through the FG models. The factors (squares) relate to the random variables (circles). As an example, in Figure 2, the factor $f_{\Delta d_{1}}$ relates the variables $\Delta d_{1}$ to Δx and Δy through the equations of Section 2.3.

### 3.1. Factor Graph with Loops

The BN (Equation 6) can be represented by a FG with the following factors (Figure 2):(7)$$f\left(x|{\underline{d}}_{1},\ldots ,{\underline{d}}_{N}\right)\propto f\left(x\right)f\left(y\right)\prod_{i=1}^{N}f\left({\underline{d}}_{i},\Delta d_{i}\right)\cdot \left(\Delta d_{i},\Delta x\right)f\left(\Delta x,x\right)f\left(\Delta d_{i},\Delta y\right)f\left(\Delta y,y\right).$$

This solution (Equation 7) has loops, which are not desirable in a FG design, because they cause indeterminate behaviors. Therefore, in the next section, a different approach that avoid loops is presented.

### 3.2. Factor Graph Avoiding Loops (or Intermediate Solution)

This section details a proposed solution that avoids loops in the factor graph. This algorithm is also named an “intermediate solution”. The random variables are grouped into vectors, and the factors can be written as follows:(8)$$f\left(x|{\underline{d}}_{1},...,{\underline{d}}_{N}\right)\propto f_{x}\left(x\right)\left(\prod_{q=1}^{Q}f_{{\underline{d}}_{q}}\left({\underline{d}}_{q},\Delta d_{q}\right)f_{\Delta d_{q}}\left(\Delta d_{q},\Delta x_{q}\right)\right)\cdot f_{\Delta x}\left(\Delta x,x\right).$$

Random variables ${\underline{d}}_{i}$ are grouped into vectors. *Q* multivariate random variables are formed with possible dimensions of D∈{N}.

Note that the computation of factors involves multivariate random Gaussian variables, and there are linear relations between them. Therefore, the designed FG can be solved with Gaussian propagation messages (Gaussian BP algorithm) [18], as explained in Appendix A. In this case, the messages can be represented by the mean vector and the covariance matrix of the corresponding multivariate Gaussian random variable.

Figure 3 shows the proposed FG, where *Q* is the result of the binomial coefficient ND. As an example, when there are ranges among tag node with N=4 different anchor nodes and also D=3 components or ranges, the ranges are grouped in Q=4 groups. Therefore, Q=4 groups of D=3 components or ranges.

### 3.3. Messages between the Components of the Factor Graph

The BP or sum-product algorithm applied to the FG allows the computation of the marginal probabilities of the random variables with the messages (μ) between the components of the FG [18]. The general rules of the sum-product algorithm are explained in Appendix A: message from variable to factor (Equation 12) and message from factor to variable (Equation 13). For the presented case, this section details the main messages between the components of the FG designed by us. The messages are shown in Figure 3, and the operations are summarized in Table 1.

For the proposed FG, normal Gaussian random variables and linear relations between variables are considered. Therefore, the integrals of (Equation 13) can be calculated directly: the marginal distribution of a joint Gaussian distribution is another Gaussian distribution, with the mean and covariance of the marginalized variable [16]. Thus, the messages can be represented by the mean and covariance of the corresponding Gaussian random variable, as shown in Table 1. In the case of a multivariate variable, the mean is a vector of mean values (Appendix A).

On the one hand, the message from a factor to a variable (Equation 13) represents the marginal of the joint distribution. On the other hand, the marginal distribution of a joint Gaussian distribution is another Gaussian distribution with a mean and covariance of the variable that is marginalized. Therefore, in messages (Table 1) such as $\mu_{f_{\Delta x}\to x}$, the marginalized variable is x and the message is calculated as $N∼(x;m_{\Delta x}+x^{0},\sigma_{\Delta x}^{2})$, where $m_{\Delta x}+x^{0}$ and $\sigma_{\Delta x}^{2}$ are the mean and covariance of x, respectively.

The message from a variable to a factor is the product of all the messages coming from the other neighboring nodes to the variable node (Equation 12). Therefore, messages (Table 1) such as $\mu_{\Delta x\to f_{\Delta d_{x}}}$ are calculated as $\prod_{q=1}^{Q}\mu_{f_{\Delta d_{q}}\to \Delta x}$.

From Figure 3, note that each message $\mu_{f_{\Delta d_{q}}\to \Delta x_{q}}$ is $N∼(\Delta x_{q};m_{\Delta x_{q}};C_{\Delta x_{q}}$); for the LS estimation case: $N∼(\Delta x_{q};{\left(H^{T}H\right)}^{-1}H^{T}m_{\Delta d_{q}},{\left(H^{T}H\right)}^{-1})\;$, and for the WLS case it is $N∼(\Delta x_{q};{\left(H^{T}WH\right)}^{-1}H^{T}Wm_{\Delta d_{q}},{\left(H^{T}WH\right)}^{-1})\;$. When D=2, H has the minimum dimensions (2 × 2) to apply pseudoinverse solving to the determined system of (Equation 3) and find a solution with LS or WLS methods, as in (Equation 4) and (Equation 5). In Section 4 of the results, N=4 ranges with D=3 components are considered, so the ranges are grouped into Q=4 sets of *D* components.

Note that the covariance of positioning with the WLS estimation $C_{\hat{x}}={\left(H^{T}WH\right)}^{-1}$ is related to the weighted geometric dilution of precision, $WGDOP=\sqrt{trace{\left(H^{T}WH\right)}^{-1})}$, of the UWB tag node related to the network of reference UWB nodes in the range [24]. In the messages $\mu_{f_{\Delta d_{q}}\to \Delta x_{q}}$ of Figure 3 and Table 1, a position solution $\Delta x_{q}$ (with LS or WLS algorithms) is obtained for each group q=1,…,Q. Message $\mu_{\Delta x\to f_{\Delta d_{x}}}=\prod_{q=1}^{Q}\mu_{f_{\Delta d_{q}}\to \Delta x}$ is a product of the incoming messages to $f_{\Delta d_{x}}$; hence, it is a product of Gaussian variables $\Delta x_{q}$, where q=1,…,Q. For this product, the mean may be
(9)$$m_{\mu_{\Delta x\to f_{\Delta d_{x}}}}=\frac{\sum_{q=1}^{Q}C_{\Delta x_{q}}^{-1}m_{\Delta x_{q}}}{\sum_{q=1}^{Q}C_{\Delta x_{q}}^{-1}}$$
and the covariance may be
(10)$$C_{\mu_{\Delta x\to f_{\Delta d_{x}}}}=\frac{\prod_{q=1}^{Q}C_{\Delta x_{q}}}{\sum_{q=1}^{Q}C_{\Delta x_{q}}}\;.$$

Thus, in (Equation 9), each solution from each group $m_{\Delta x_{q}}$ obtained with *D* ranges is weighted with its covariance $C_{\Delta x_{q}}$ to obtain the solution Δx. This allows groups with better covariance results (related to WGDOP for the WLS case) to contribute more than the others to estimating the variable Δx, and thus the position x.

Upward messages may be calculated following the described rules. One iteration of the algorithm ends once two messages have been passed over each edge, one in each direction. Then, the marginal probabilities of the random variables can be estimated as the product of the incoming messages (Appendix A). For x, the marginal probability may be the product of the following messages toward x, which are shown in Figure 3 and Table 1: $x=\mu_{f_{\Delta x}\to x}\cdot \mu_{f_{x}\to x}$ .

This is a product of normal Gaussian functions that allow them to be weighted according to the corresponding covariance.

The iterations are carried out until the algorithm results converge to a solution. This is explained in the pseudo-code (Algorithm 1) of the algorithm avoiding loops that is detailed in next Section 3.4.

### 3.4. Pseudocode of a Factor Graph-Based Algorithm That Avoids Loops

The main steps to solve the factor graph model that avoids loops (intermediate solution) (Figure 3) with the Gaussian BP algorithm are detailed in Algorithm 1. As explained in Section 3.3, the algorithm considers LS or WLS techniques for the estimation of the message $\mu_{f_{\Delta d_{q}}\to \Delta x_{q}}$. In the first step of the algorithm, the initialization of variables is performed. There are variables related to the scenario, FG models, and Taylor series. Then, the FG is solved with the BP algorithm performing iterations until reaching convergence. Note that the Taylor series equation was introduced in Section 2.

As explained previously, the tag and anchor commercial nodes are IEEE 802.15.4 compliant with the UWB-based physical layer. The communications of the UWB nodes are organized following the superframe structure for time division duplexing (TDD). Thus, the tag node knows the anchor node positions in this range. The ranging protocol between the tag and the corresponding anchor nodes provides these positions to the tag. The superframe structure and the ranging protocol are described in the standard IEEE802.15.4 [21] (and previous versions). Therefore, it is considered that the position variable of the tag, x, is initialized as $x_{\mathrm{init}}^{0}=\frac{\sum_{i=1}^{N}x_{a_{i}}}{N}$ and covariance $\sigma_{x_{\mathrm{init}}^{0}}^{2}$$=\left(\begin{array}{cc} 100 & 0\\ 0 & 100 \end{array}\right)$ m., for the first iteration of the algorithm.

For each iteration of the algorithm, the input data of the algorithm from the dataset are the same for a given scenario:${\underline{d}}_{i}$ are the distances from the tag to the anchor nodes, and $x_{a_{\left(i\right)}}$ are the anchor node coordinates;Variables $\sigma_{{\underline{d}}_{i}}^{2}$ are initialized with statistics from the distances of the dataset.

In each iteration, down and up message passing and estimations are performed (Figure 3). After each iteration, a marginal probability of the tag position x and its covariance $\sigma_{x}^{2}$ is estimated and updated through the initialization of variables. In addition, variables ${(}^{0})$ of Taylor algorithm are updated through initialization (see Equation (Equation 2)).
**Algorithm 1** Factor graph avoiding loops (Figure 3)   1:Initialization of variables of the scenario: $x_{a}$,$\underline{d_{i}}$,$\sigma_{d_{i}}^{2}$;   2:Initialization of variables of the algorithms: Taylor series and FG ($\underline{d_{i}}$, x);   3:k = 1;   4:**while** k ≤ iterations **do**   5:    Down messages (1) of BP: Δd for each group; Δx with LS (4) or WLS (5); and x   6:    Up messages (1) of BP: x;   7:    **if** Termination **then**   8:        Result (Marginal functions): x and $\sigma_{x}^{2}$.   9:        Variables of FG and Taylor algorithm are updated for initialization. 10:    **end if** 11:    k = k + 1; 12:**end while**

In the next Section 4, the results are shown. In the figures, the results obtained are compared to the four solutions: *(Sol.A)* iterative FG algorithm to avoid loops (Algorithm 1) with the WLS algorithm to solve messages $\mu_{f_{\Delta d_{q}}\to \Delta x_{q}}$ for each group q=1,…,Q (Section 3.3). This solution is named in Figures as “WLS with FG-Intermediate”; *(Sol.B)* iterative FG algorithm avoiding loops (Algorithm 1) with the LS algorithm to solve messages $\mu_{f_{\Delta d_{q}}\to \Delta x_{q}}$ for each group q=1,…,Q (Section 3.3). This solution is named in Figures as “LS with FG-Intermediate”; *(Sol.C)* iterative classical LS and WLS algorithms (Section 2.3); and *(Sol.D)* positioning data collected from UWB commercial nodes and stored in published datasets [23]. In the case of the iterative approaches *(Sol.A)*, *(Sol.B)*, and *(Sol.C)*, the initialization and updating of variables (related to the scenario and variables of the Taylor series algorithm in each iteration) follows the same procedure as was explained in Algorithm 1. However, for *(Sol.C)*, the variable obtained x in each iteration is used to update the Taylor algorithm for initialization. Note that in this case, x is the result with classical approaches, so it is not the marginal function. Moreover, the classical algorithms cannot estimate the covariance of the position $\sigma_{x}^{2}$ in its equations of positioning as the FG does.

## 4. Results and Discussion

This section shows the results obtained with the approaches presented previously. To denote the name of the algorithms in the following figures, the names of the solutions *(Sol.A)*, *(Sol.B)*, and *(Sol.C)* were given in Section 3.4. The results with solution *(Sol.D)* are also shown in this section. Thus, in the figures, on the one hand, the LS and WLS models with FG without loops (intermediate solution) are named “LS or WLS with FG - Intermediate” and, on the other hand, LS and WLS are iterative classical algorithms without the FG technique. The presented algorithms were tested with simulated data and also with real-life data collected in scenarios with one target node, three or four anchor nodes, and the TOA-based range technique.

For *(Sol.A)* and *(Sol.B)*, D=3 is assumed (Section 3.2); therefore, it utilized groups of 3 ranges. The ranges were grouped in Q=4 groups of D=3 components or ranges. For classical approaches, the algorithms presented in Section 2.3 were considered. The results of the algorithms were averaged over $N_{C}=1000$ independent Monte Carlo trials. The root mean square error (RMSE) of the target position was estimated with the following equation:(11)$$RMSE=\sqrt{\frac{\sum_{\ell =1}^{N_{C}}{∥{\hat{x}}^{\left(\ell \right)}-x_{\mathrm{TAG}}∥}^{2}}{N_{C}}}\;.$$

In (Equation 11), in the case of results with collected data, $x_{\mathrm{TAG}}$ is the calculated position of the tag node with a total station and ${\hat{x}}^{\left(\ell \right)}$ is the estimated position with the different presented methods. Moreover, the positions of the anchor nodes were also calculated with a total station [23].

### 4.1. Simulation Results

In this case, the algorithms were tested with simulated data generated in a good scenario in terms of positioning: the anchor nodes were placed forming a good geometry and they were in good transmission conditions (LOS). Ranges were modeled with the ranging model presented in (Equation 1).

The results with the simulated data show the behavior of the FG based algorithm in the good scenario. UWB technology is robust against ranging and positioning errors due to transmission conditions such as multipath (without severe NLOS conditions). Hence, the parameters $\sigma_{{\underline{d}}_{i}}^{2}$ for the ranges were set with low values, similarly to in the good scenario B (Section 4.2.4). A conceptual representation of the concept behind the WGDOP metric is shown in Figure 1. The geometrical situation of the anchor nodes affects the positioning error, which is quantified using the WGDOP. This issue is shown in Figure 4, where in a scenario as a grid, the algorithms results are represented every 0.5 m, thus showing the geometrical effect on the positioning error. There are small differences between the RMSE of FG and RMSE of WLS classical algorithm in the central part of the figure, shaped like a cross (red color). In this area, geometrical issues did not affect the RMSE result. However, the RMSE of WLS algorithm increased from this part towards the corners of the scenario. Thus, the difference between the RMSE of the WLS algorithm and the RMSE of the FG-based algorithm increased towards the corners. This geometrical effect that increased the RMSE value could be corrected by the FG-based algorithm.

In Section 4.2, the tests with real collected data showed that the FG-based algorithm takes into account both the effect of the distance error from the tag to the anchor nodes and the geometry conditions on the positioning solution.

### 4.2. Results with Real Collected Data

The results with collected real data were obtained with measurements of the published datasets in the Zenodo repository: “Datasets of Indoor UWB Measurements for Ranging and Positioning in Good and Challenging Scenarios” [23], which were collected from commercial UWB devices. In Section 4.2.1, the measurement environment, the measurement setup, and the data collection are described. In Section 4.2.2, the scenarios are detailed, and in Section 4.2.3, Section 4.2.4 and Section 4.2.5, the results for the collected data in challenging, good, and intermediate scenarios are explained. The results section ends with some remarks in Section 4.3.

#### 4.2.1. Datasets of Collected Data

The “Datasets of Indoor UWB Measurements for Ranging and Positioning in Good and Challenging Scenarios” [23] are available at the Zenodo repository. The datasets were collected at CTTC’s Indoor Navigation Laboratory [25]. Range and positioning related measurements were collected from the MDEK1001 system, which consists of DWM1001 UWB modules (DW) [15] from Decawave (Qorvo). Nodes contain the DW1000 chip that is IEEE 802.15.4 [21] (UWB physical layer) standard compliant, with the following configuration: channel 5–6.5 GHz, 6.8 Mbps, and 500 MHz bandwith.

This real-time location system (RTLS) based on UWB is located in the indoor navigation lab. The lab is a static indoor laboratory environment, as shown in Figure 5. This RTLS is based on anchor-based positioning. First, the tag node estimates the range between tag and the corresponding anchor nodes. Second, the tag node estimates the position and quality factor of the position. The tag node reports the position and quality factor if applicable. This type of positioning is affected by parameters such as propagation conditions and the geometry formed by the anchor nodes. The positioning algorithm of the manufacturer implemented on the tag UWB node (DW) is based on the maximum likelihood method.

During data capture, there may be systematic errors due to the setup of the scenarios or the hardware used or other causes. Thus, data were collected at different dates and times and with different setups of the same scenarios. However, a typical location accuracy is around X − Y < 10 cm in LOS, following the specifications of the manufacturer. More details about the measurement setup and nodes configuration are provided in [23].

Anchor nodes (red triangles) were placed on the walls, with the tag node (black dot) on a tripod. Anchor and tag nodes were placed in reference positions, whose coordinates were estimated with a total station. This static indoor lab environment allowed setting up scenarios and changing the main conditions that affect the positioning performance: different propagation conditions and different geometries. Thus, the scenario setups included LOS and non-LOS propagation conditions, as well as easy, medium, and challenging geometries.

The UWB DW nodes were mounted (i) at different positions in the laboratory with different geometries and (ii) with different propagation conditions (e.g., multipath and NLOS conditions) that affected the distance error. LOS conditions were considered with no obstacles between the corresponding anchor and tag nodes. *softNLOS* and *hardNLOS* conditions were configured with moving obstacles (e.g., obstacles made of cardboard and metal, respectively) between the corresponding anchor node and the tag node in a controlled manner. In the *softNLOS* condition, the standard deviation of the range error increased and a small bias could be included. Under the *hardNLOS* condition, the range error had a higher bias and the standard deviation of the range error also increased. In this case, the bias was caused by the obstacles between the corresponding anchor and the tag.

The following data were collected on a laptop from the USB port of the tag node:Distances to anchor nodes: the update rate of the DWM1001 systems was set to 10 Hz;Position of the tag node: this was collected from the USB of the tag node. The location update rate was set to 10 Hz. Position was estimated by the tag node when three conditions were met: (i) tag node had three or more tag-anchor distances estimated; (ii) internal location engine (LE) of the tag node was enabled. LE reported the position and quality factor; and (iii) positions of anchor nodes had to be stored in the memory of the anchor node;Position quality factor: this is a parameter whose value ranges from 0 to 100, with a value close to 100 indicative of good positional quality. More details can be found in the datasheet of the DWM1001 device and in the published dataset description [23].

#### 4.2.2. Scenario Settings

The results of the algorithms with approaches “LS or WLS with FG - Intermediate” were compared with the results of the classical iterated LS or WLS approaches and with the results of the UWB commercial system DW. The results presented in the next sections were obtained with input data from the following scenarios of the dataset [23]: benign scenario (Scenario B, Section 4.2.4), intermediate (Scenarios C1 and C2; Section 4.2.5), and challenging scenarios (Scenarios A1, A2, A3, and A4; Section 4.2.3). The scenarios are shown in Figure 6 and the tables: Table 2 shows the propagation conditions and geometry of the nodes placement, while Table 3 shows the scenario settings related to the distance error. For each scenario, a laboratory map is presented with the placement of the anchor and tag nodes, the WGDOP results for each group, and RMSE results estimated with (Equation 11). Note that in the figures showing the RMSE results, iteration 1 means RMSE results with initialization values (tag position and other variables) for each iterative algorithm.

The parameters $w_{i}$ were set from the real data collected from the commercial UWB nodes of the dataset [23]. Thus, $w_{i}$ was estimated with statistics ($w_{i}=\sigma_{{\underline{d}}_{i}}^{-2}$). Usually, the range between two UWB nodes such as the anchor *i* and the tag node follows a Gaussian distribution. $\sigma_{{\underline{d}}_{i}}^{-2}$ were estimated with post-processing. In a future work, we will study this estimation more in detail. Moreover, with real data, the improvement and the effect on the position estimation of the grouping of variables was studied. In this way, the WGDOP value estimated by the tag node was presented for each group and for each scenario. The estimated position of the group with the best WGDOP value had the highest weight in the positioning solution with the FG-based algorithm with WLS. The WGDOP was related to the covariance result of each group $C_{\Delta x_{q}}$ with the WLS method, as explained in Section 3.3.

#### 4.2.3. Results: Challenging Scenarios

• Scenario A1: This was a challenging scenario for positioning purposes that included three anchor nodes, in *hardNLOS* conditions placed with a good geometry. In Figure 7, histograms of ranges have bias, an increased standard deviation, and there is a histogram of a node that does not follow the Gaussian distribution curve. Scenario A1 is shown in Figure 6, while the estimated RMSE is shown in Figure 8. There was 1 group with good WGDOP because the geometry was good. The WLS results were better than the LS results. The aim of this scenario was to show that the algorithm FG with WLS (named “WLS with FG-Intermediate” in the figures) assigned more weight to the ranges with less error, similarly to the WLS algorithm. The RMSE of the DW position X-Y (m) (0.1283) and RMSE (m) of FG-WLS and WLS solutions had similar values.

• Scenario A2: This was a challenging scenario that included four anchor nodes, two nodes in LOS and two in *hardNLOS* conditions placed with medium geometry. In Figure 9, histograms of ranges of nodes with NLOS conditions have a higher bias and increased standard deviation, whereas the histogram of nodes in the LOS condition follows the Gaussian distribution curve. Scenario A2 is shown in Figure 6, whereas the estimated WGDOP and RMSE are shown in Figure 10 and Figure 11, respectively. The group with best WGDOP (in iteration 10) was the second one (WALL1c,1d,7). Although the $\sigma_{d_{i}}$ of node WALL 7 was higher than for node 1b, WALL 7 improved on WGDOP in terms of geometry. This was a scenario with high positioning errors, but the algorithms did not diverge. Iterations reduced the error, and the RMSE of WLS with FG was 0.1 m (iteration 4), which was lower than the RMSE of 0.9815 m provided by the DW nodes.

• Scenario A3: This was a challenging scenario that included three anchor nodes in *hardNLOS* propagation conditions placed with medium geometry. In Figure 12, histograms of ranges have bias, an increased standard deviation, and there is a histogram that does not follow the Gaussian distribution curve. Scenario A3 is shown in Figure 6, whereas the estimated WGDOP and RMSE are shown in Figure 13, Figure 14 (11 iterations) and Figure 15 (500 iterations) respectively. In this scenario the tag position was estimated with high error, but algorithms still did not diverge. It can be seen that the algorithm iterations reduced the error, but the FG algorithm with WLS needed more iterations than the WLS to converge. However, it converged to a solution with the minimum positioning error.

• Scenario A4: This was a challenging scenario that included four anchor nodes: one node in LOS and three nodes in *hardNLOS* conditions placed with challenging geometry. In this scenario, there were many distances from the anchor initial node (WALL 1c) that the tag node did not estimate. In these cases, the DW UWB tag did not report the position. This was due to the challenging geometry and NLOS conditions of the UWB nodes that formed the UWB network. These conditions affected, on the one hand, the communications between the nodes that formed the UWB network and the range protocol and, on the other hand, the estimation of distance and position. Scenario A4 is shown in Figure 6, whereas the estimated WGDOP, RMSE and RMSE (zoom) are shown in Figure 16, Figure 17 and Figure 18, respectively. The WGDOP value was estimated by the tag node in each iteration. This was a scenario with a high RMSE error of position, in which the classical algorithms diverged (LS and WLS) but FG-based ones with LS and WLS did not. Iterations reduced the error. Group 2 (WALL1c, WALL1d, WALL1a) had the best WGDOP values; therefore, the algorithm assigned more weight to the result $m_{\Delta x_{q}}$ of this group. The RMSE of the position X-Y with DW system was 0.9730 m, and the RMSE of the position for FG with WLS was around 0.9 m (iteration 4) and 0.55 m (iteration 8).

#### 4.2.4. Results: Good Scenarios

• Scenario B: This was a good scenario, in terms of the geometry and propagation conditions. The number of anchor nodes was four, and the WGDOP values of each group were good. Scenario B is shown in Figure 6, whereas the estimated WGDOP and RMSE are shown in Figure 19 and Figure 20, respectively. The results of the algorithms without FG did not diverge, they were similar to the results of the WLS algorithm with FG. The WGDOP value of group 2 (nodes WALL1c, WALL3 and WALL6) was lower than the others. Furthermore, in LOS conditions, the histogram of the distances between each anchor node and the tag followed a normal distribution curve. If the errors follow a normal distribution and the model is linear, the LS based estimators are also the maximum likelihood estimators. This is shown in the RMSE results. The RMSE results were similar: the RMSE values of algorithms with FG, RMSE of algorithms with LS and WLS, and DW RMSE Position X-Y (0.0172 m). In addition to scenario B, the RMSE results (Figure 21) for scenario B(2) are presented. In scenario B(2) the position of the tag was (3.277,3.2934) m different than in B, that is (3.277,4.2934) m, thus it was shown that the FG algorithm with WLS converged regardless of the position of the tag.

Figure 4 shows the geometrical effect on the positioning error in a good scenario, with good propagation conditions, and with simulated results. It can be observed that the geometrical effect that increased RMSE values in some parts of the scenario could be corrected by the FG-based algorithm.

#### 4.2.5. Results: Intermediate Scenarios

• Scenario C1: This was an intermediate scenario with four anchor nodes in *softNLOS* with intermediate geometry. Scenario C1 is shown in Figure 6, whereas the estimated WGDOP and RMSE are shown in Figure 22 and Figure 23, respectively. The WGDOP value of group 2 (WALL1c,1d,7) was the lowest at iteration 10. Note that for WALL7 the value $\sigma_{{\underline{d}}_{i}}$ (m) was higher than for the other anchor nodes but its position improved the geometry issues with the WGDOP value. In *softNLOS* conditions, the error $\sigma_{d_{i}}$ was increased in the error distribution, but following Gaussian issues without high bias. Therefore, the results of the algorithms with FG were similar to the results of algorithms LS and WLS. Moreover, the RMSE value of the algorithm with FG and WLS was 0.06 m (iteration 4) and the RMSE value with DW system was 0.0559 m, proving similar.

• Scenario C2: This was an intermediate scenario with four anchor nodes in *softNLOS* with poor geometry. Scenario C2 is shown in Figure 6, whereas the estimated histogram of ranges, WGDOP and RMSE are shown in Figure 24, Figure 25 and Figure 26, respectively. The WGDOP value of the group 2 (WALL1c,1a and 1b) was the lowest at iteration 10. Therefore, the value of their weight was higher and $\sigma_{{\underline{d}}_{i}}$ (m) was lowest. The RMSE values of the non-FG-based algorithms were derived, while the FG-based algorithms converged to a solution with a value similar to the DW RMSE value.

### 4.3. End Remarks

The convergence of the belief propagation algorithm to the optimal solution is shown in the results for all the different scenarios. As we commented previously, the convergence was ensured, since the FG was modeled as a tree (without loops).

The state of the art of Section 1 introduced approaches based on WLS and DW that take into account both the effect of propagation conditions and the effect of geometry on anchor-based positioning. In this Section 4, results with the proposed solutions were obtained and compared with the results of the state-of-the-art approaches. On the one hand, the results with simulated data in Section 4.1 showed that the geometrical situation of the anchor nodes affected the positioning error. This geometrical effect that increased the RMSE value could be corrected by the FG-based algorithm, but it was not corrected by the WLS-based algorithm. On the other hand, in the results with real collected data in Section 4.2, the proposed solutions were compared with the introduced state-of-the-art approaches for all the presented scenarios. In the case of scenarios in which the WGDOP of the tag node was poor, due to poor geometry and poor propagation conditions (that affected the distance estimation), the presented iterative FG without loops achieved better results than the iterative LS and WLS algorithms. The FG without loops allowed the weighting of the group with the best WGDOP. This meant the positioning results for the algorithm based on FG without loops were better than those estimated with algorithms not based on FG, such as the LS-based ones in challenging scenarios, when the WGDOP of the target node was poor. Moreover, the presented FG without loops outperformed the DW solution in very challenging scenarios.

## 5. Methods

The algorithms of Section 4 (Results) were programmed and tested with the programming and numeric computing platform Matlab R2019b. The pseudocode of the algorithms code, as well as the simulation parameters, are detailed in Section 3.4. The computer code can be requested from the corresponding author.

Regarding the input data of the algorithms, the input simulated data are detailed in Section 4.1.The real-life collected data, the measurement environment, and laboratory scenario are explained in Section 4.2 (results with real collected data) and in the published dataset that is available from Zenodo [23].

The parameters (i.e. standard deviation) used to obtain the results with real collected data (Section 4.2) were estimated with post-processing of the collected data of the dataset. The scenarios of this dataset were designed taking into account the statistical parameters of the corresponding range anchor tag and the geometrical situation of the anchor nodes. The statistical parameters were the histogram, the shape of the Gaussian distribution, standard deviation, mean, bias, and so on. Thus, by modifying (i) the propagation effects such as LOS, softNLOS, hardNLOS with cardboard or metal obstacles, and (ii) the geometrical situation of the anchor nodes, the desired conditions of real-life but controlled scenarios were achieved.

## 6. Conclusions

This work presented factor graph models built on Bayesian inference in Bayesian networks, using a belief propagation algorithm and designed for hybrid and cooperative positioning. The authors described linear systems that consist of factor graphs (with Gaussian distributions) for determining the probability of the target position given the distance measurements to the anchor nodes that know the position. Thus, by using this framework, a data-driven factor graph model was designed, to perform anchor-based positioning. One of the key objectives was to design a hybrid structure (that involves data and model approaches) to address the models for positioning from a Bayesian point of view, customizing them for each technology and scenario.

The convergence of the belief propagation algorithm to an optimal solution is shown in the results. The convergence was ensured, since the FG was modeled as a tree (without loops). Moreover, the FG was a linear model with Gaussian BP (Gaussian aleatory variables), which simplified the calculation of the integrals of the messages between the factors and variables that formed the FG.

The performance results of anchor-based positioning depend on the propagation conditions between nodes and also the anchor nodes’ position. Therefore, the weighted geometric dilution of precision (WGDOP) metric, which measures the effect on the positioning solution of distance error to the corresponding anchor node and network geometry of anchor nodes, was taken into account.

Another objective of this work was to study techniques based on machine and deep learning, to improve positioning in good, intermediate, and challenging scenarios. In the presented case, anchor-based positioning techniques that used radio devices, such as (UWB, IEEE 802.15.4) were considered.

In order to test the algorithms, this work used both simulated and real-life data from a published dataset that was collected in the laboratory from UWB commercial nodes. The results showed that the presented algorithm based on factor graphs took advantage of weighted techniques. It allowed distances with better covariance results to contribute more than the others to the final position estimation. Moreover, the proposed algorithm grouped distances (random variables) to the anchor nodes, on the one hand, to avoid loops in the FG and, on the other hand, so that the position solution of each group was weighted based on its covariance. The covariance results of each group were related with the metric WGDOP. Therefore, the presented factor graph solution considered position estimations depending on the covariance results and taking advantage of Bayesian inference. The results showed that it is an iterative algorithm that converges to a solution, outperforming the classical algorithms and UWB commercial systems. The presented factor graph algorithm for anchor-based positioning takes into account both, the effect on the positioning solution of distance error on the corresponding anchor node and the network geometry of the anchor nodes.

In future works, FG modeling will be considered with techniques based on other ranging and positioning methods and other technologies, taking into account conditions such as the propagation conditions, geometry, and so on that could affect the positioning results.

## Figures and Tables

**Figure 1 sensors-23-05660-f001:**
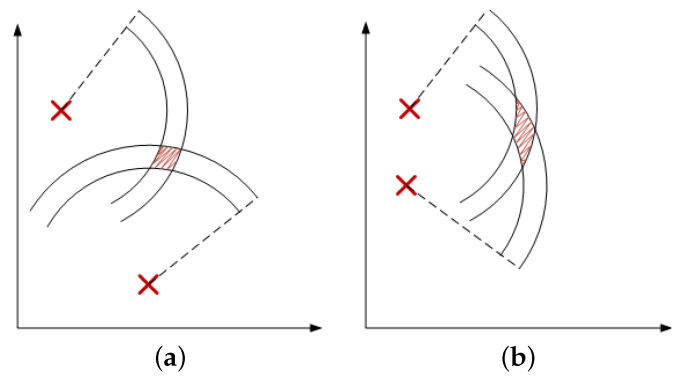
Conceptual representation of good and bad two-dimensional geometries, (**a**) and (**b**), respectively [20].

**Figure 2 sensors-23-05660-f002:**
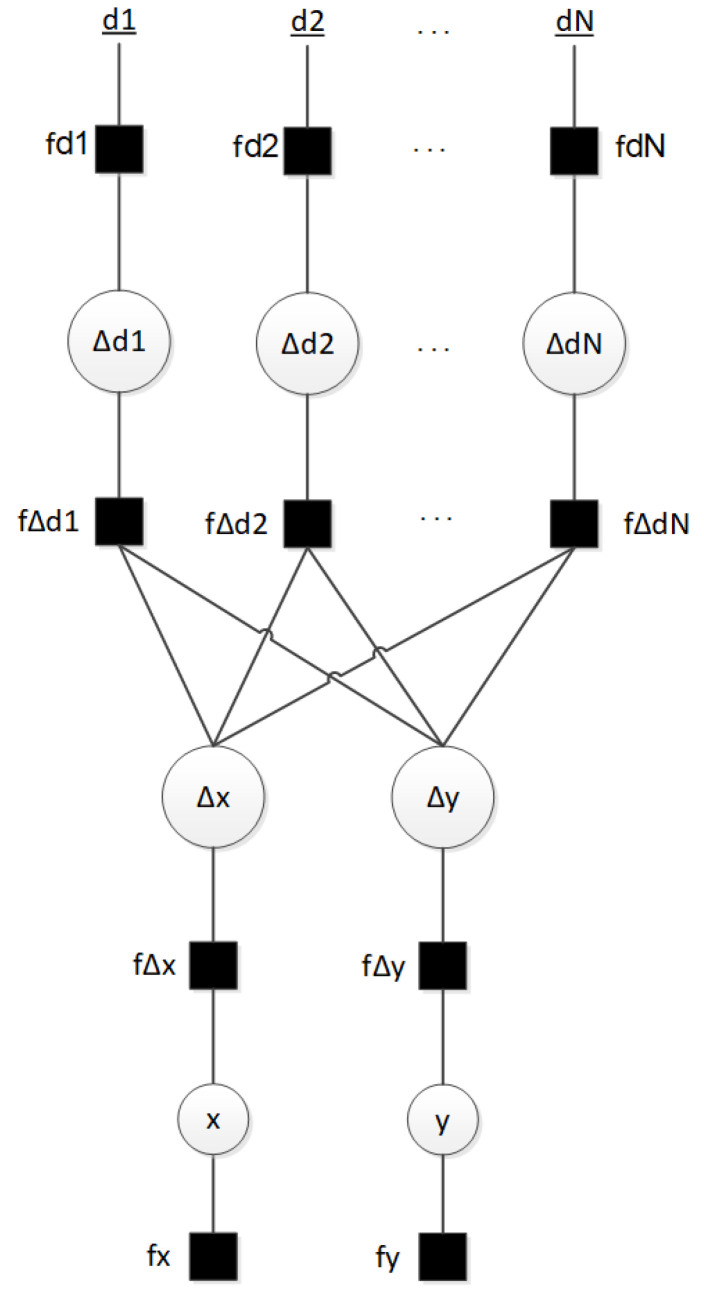
Factor Graph with loops.

**Figure 3 sensors-23-05660-f003:**
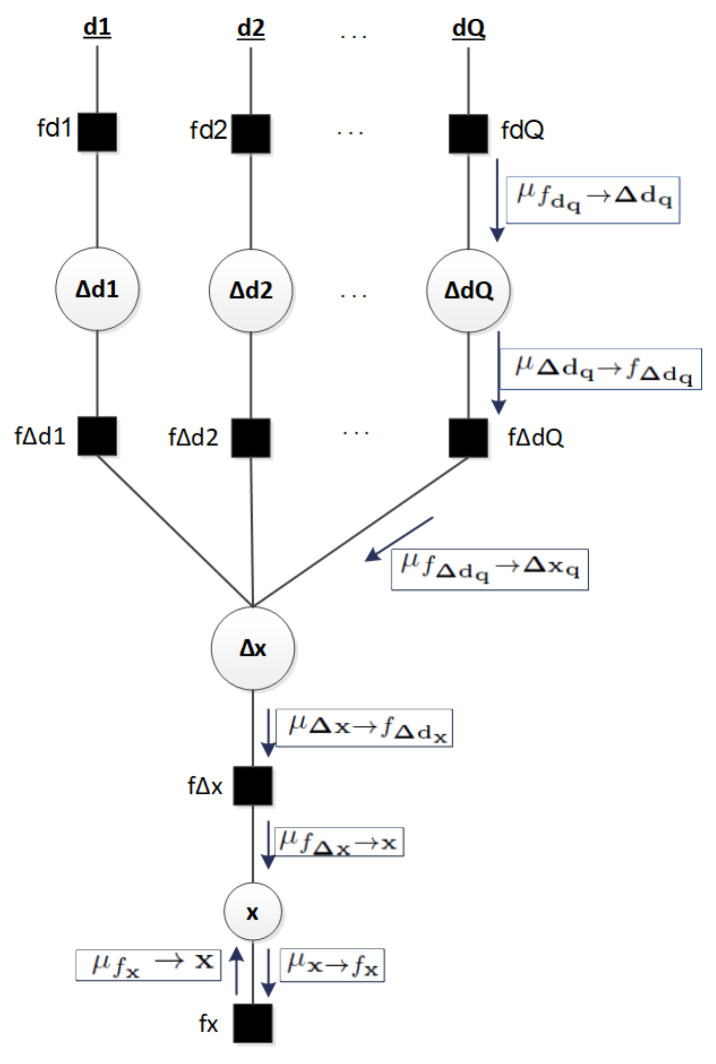
FG avoiding loops with the main messages of the BP algorithm.

**Figure 4 sensors-23-05660-f004:**
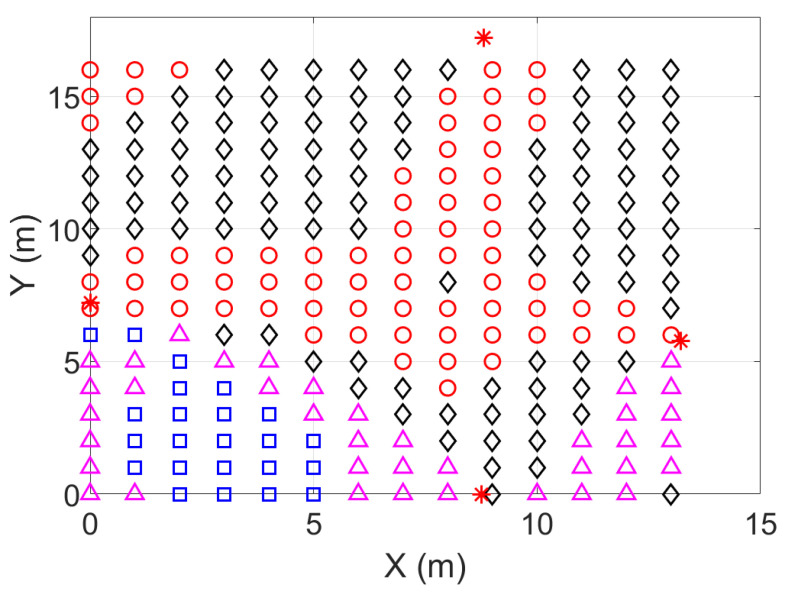
Scenario with simulated positioning results (RMSE) of the algorithms. The geometrical situation of the anchor nodes affected the positioning error. The RMSE of positioning with the classical approach (WLS) increased towards the corners. This geometrical effect could be mitigated by the FG-based algorithm. Legend: * Anchor node placement (non-star shape is the placement of the tag node); ∘ RMSE (of WLS) ≤ RMSE (of FG algorithm); ⋄ (RMSE (WLS) – RMSE (FG)) < 0.02 m; ▵
0.02 m < (RMSE (WLS) – RMSE (FG)) < 0.04 m; □ (RMSE (WLS) – RMSE (FG))>0.04 m.

**Figure 5 sensors-23-05660-f005:**
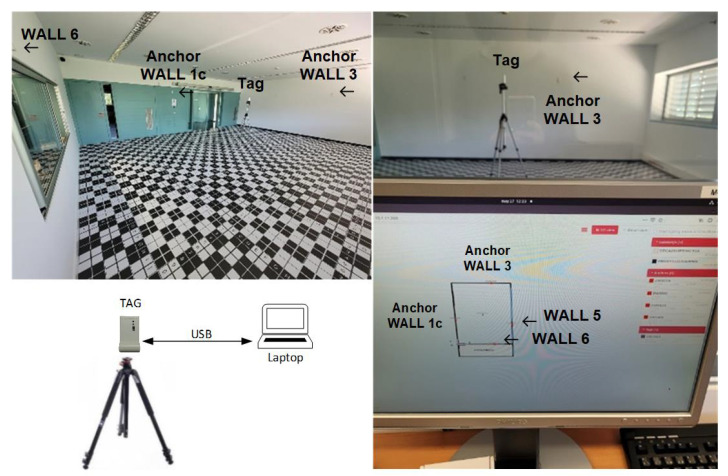
Dataset was collected at the Indoor Navigation Laboratory of the CTTC [25]. The pictures show the lab, the tag UWB node placed on a tripod, anchor nodes placed on the walls, and a map of the lab (in this case, the scenario is Scenario B).

**Figure 6 sensors-23-05660-f006:**
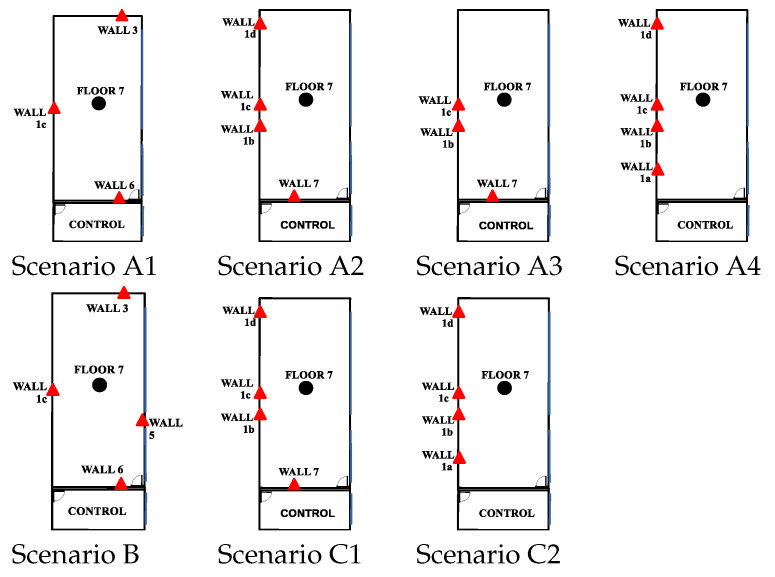
Scenarios from [23] with node placements in the map of the lab. The size of the lab is (X ∈{0, 6.607}, Y ∈{0, 8.605} m). The red triangles are the anchor UWB nodes and the black dot is the target UWB node. Table 2 shows the propagation and geometry conditions of each anchor node in the scenarios. Different NLOS conditions were configured with moving obstacles (e.g., obstacles made of cardboard and metal).

**Figure 7 sensors-23-05660-f007:**
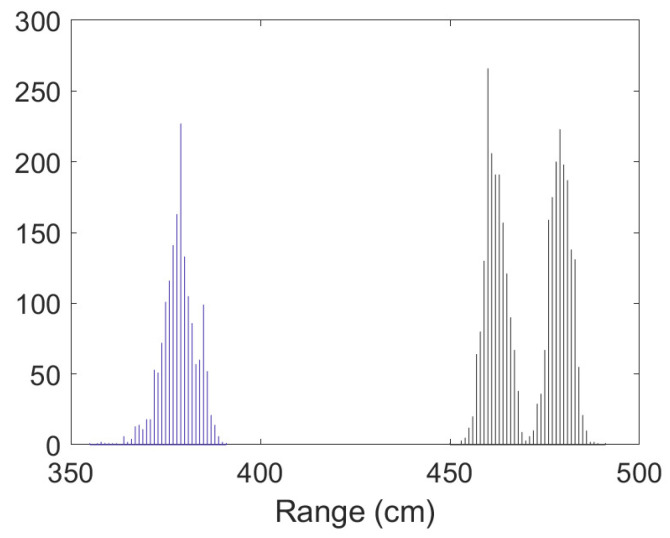
Scenario A1: Histogram of ranges.

**Figure 8 sensors-23-05660-f008:**
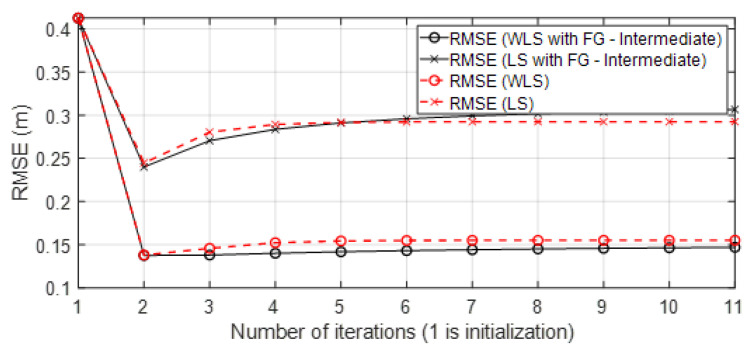
Scenario A1: RMSE.

**Figure 9 sensors-23-05660-f009:**
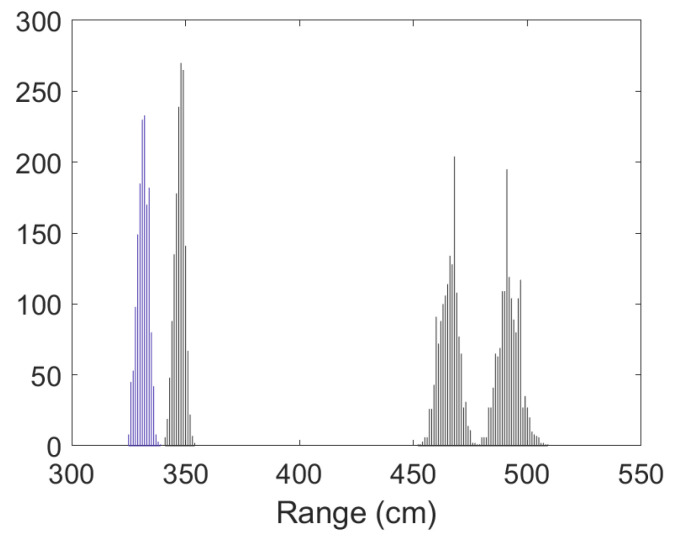
Scenario A2: Histogram of ranges.

**Figure 10 sensors-23-05660-f010:**
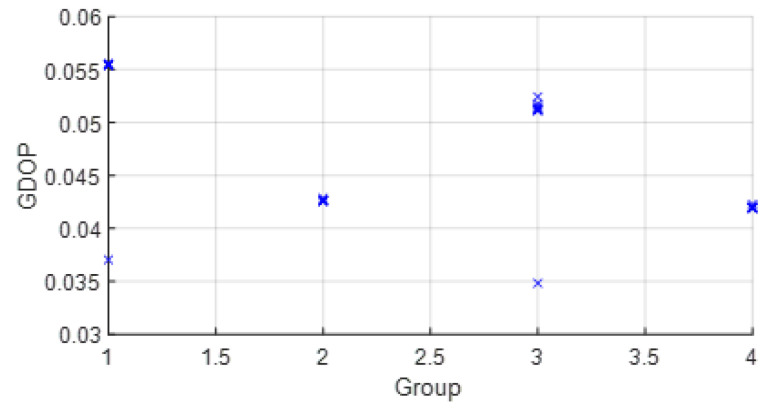
Scenario A2: WGDOP for each group of distances.

**Figure 11 sensors-23-05660-f011:**
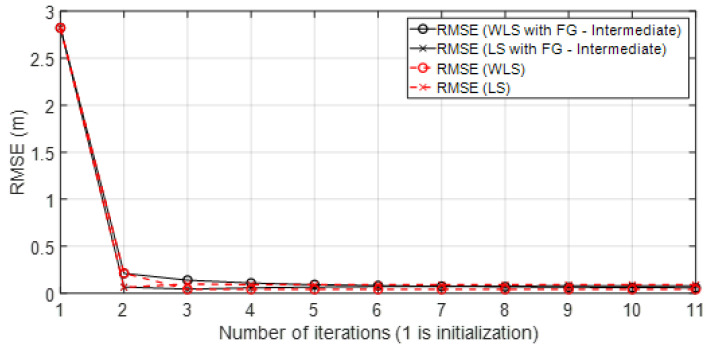
Scenario A2: RMSE.

**Figure 12 sensors-23-05660-f012:**
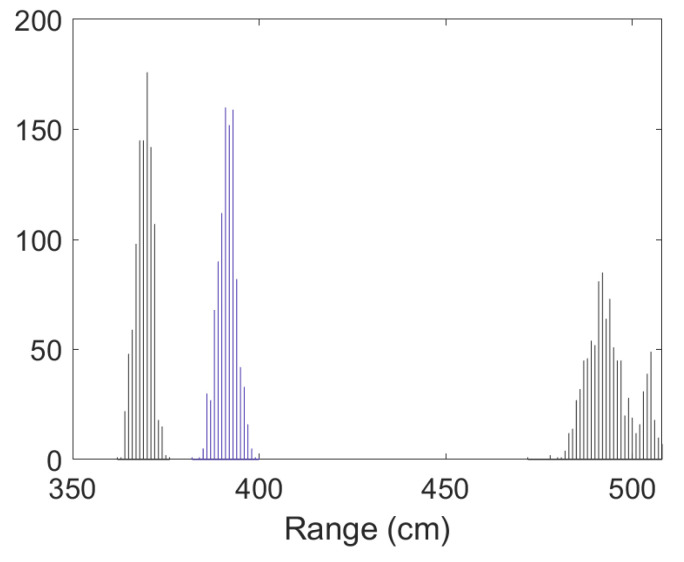
Scenario A3: Histogram of ranges.

**Figure 13 sensors-23-05660-f013:**
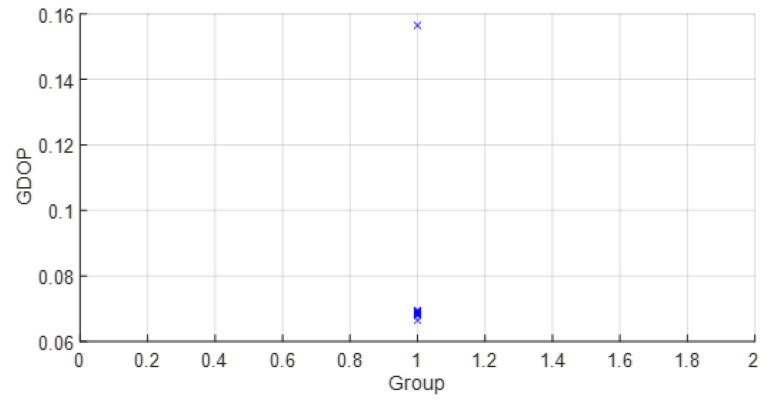
Scenario A3: WGDOP for each group of distances.

**Figure 14 sensors-23-05660-f014:**
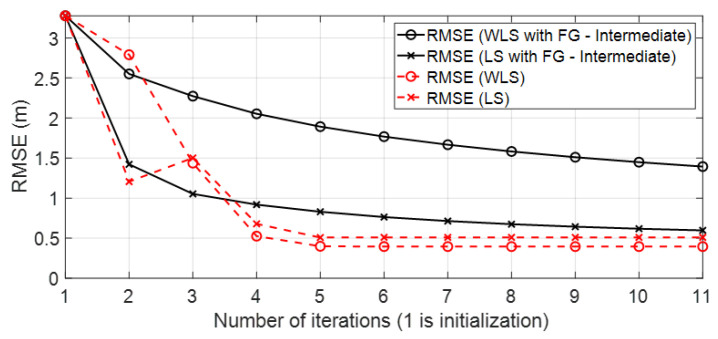
Scenario A3: RMSE for first iterations.

**Figure 15 sensors-23-05660-f015:**
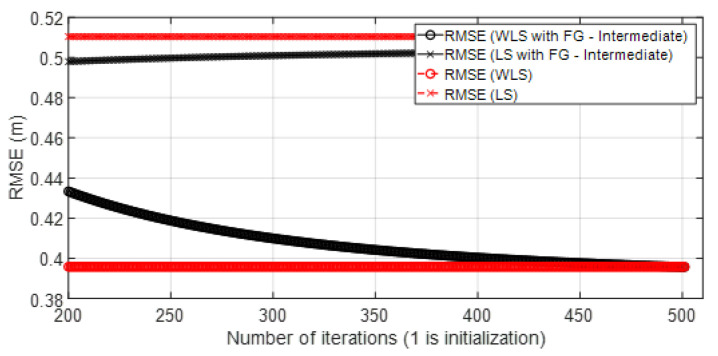
Scenario A3: RMSE for 500 iterations.

**Figure 16 sensors-23-05660-f016:**
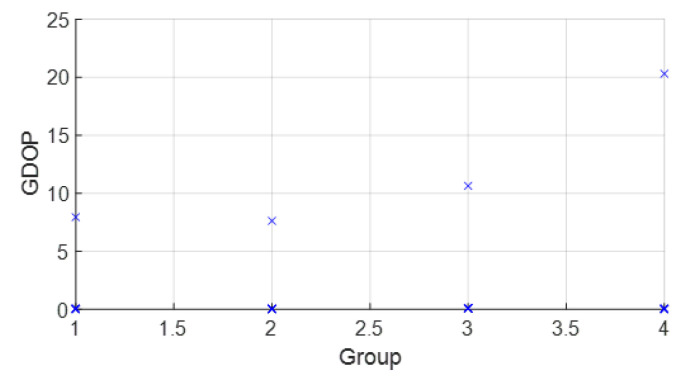
Scenario A4: WGDOP for each group of distances.

**Figure 17 sensors-23-05660-f017:**
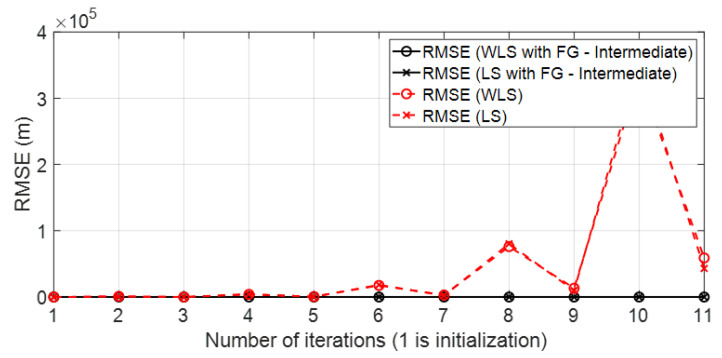
Scenario A4: RMSE.

**Figure 18 sensors-23-05660-f018:**
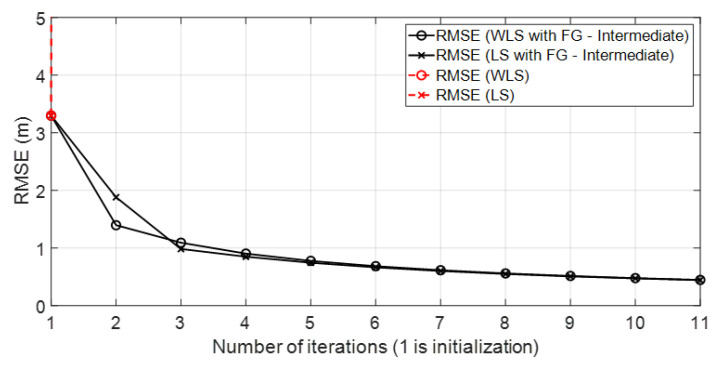
Scenario A4: RMSE (zoom).

**Figure 19 sensors-23-05660-f019:**
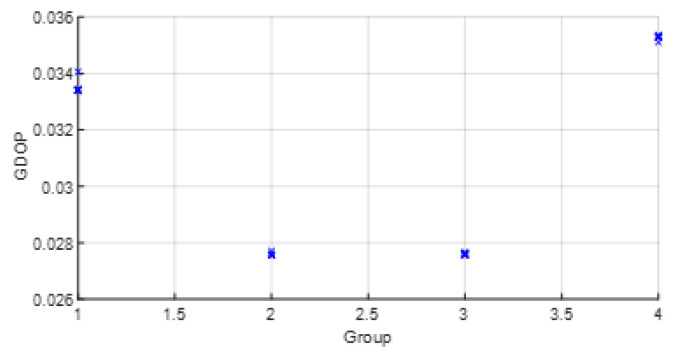
Scenario B: WGDOP for each group of distances.

**Figure 20 sensors-23-05660-f020:**
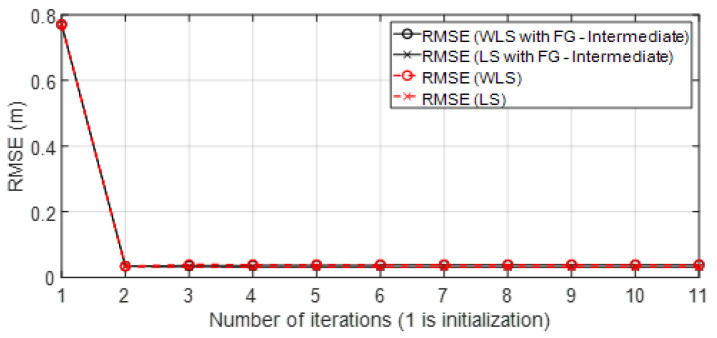
Scenario B: RMSE.

**Figure 21 sensors-23-05660-f021:**
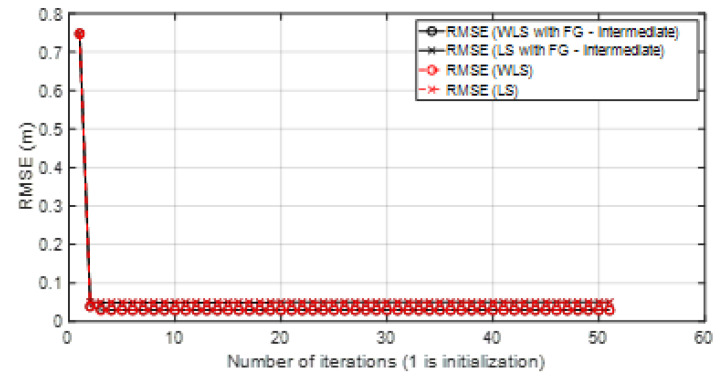
Scenario B(2): RMSE for scenario B(2).

**Figure 22 sensors-23-05660-f022:**
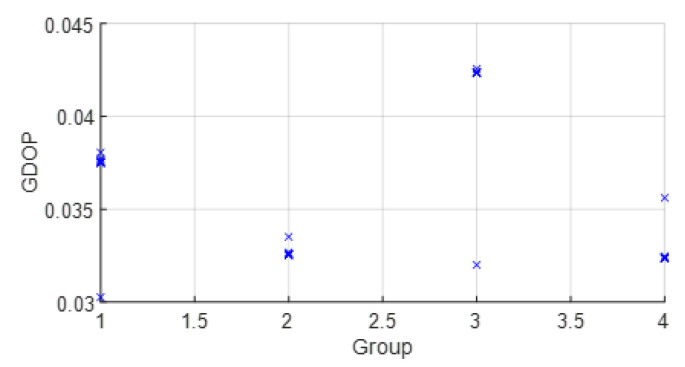
Scenario C1: WGDOP for each group of distances.

**Figure 23 sensors-23-05660-f023:**
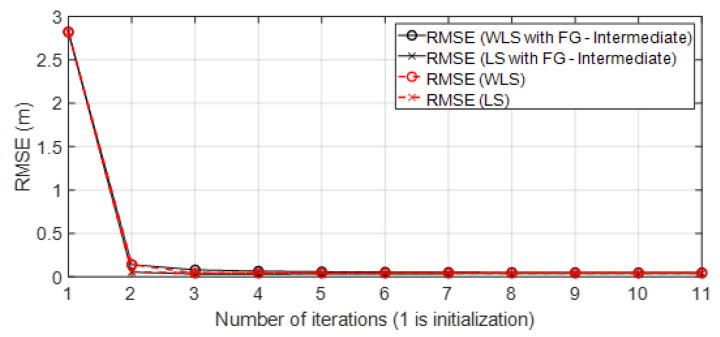
Scenario C1: RMSE.

**Figure 24 sensors-23-05660-f024:**
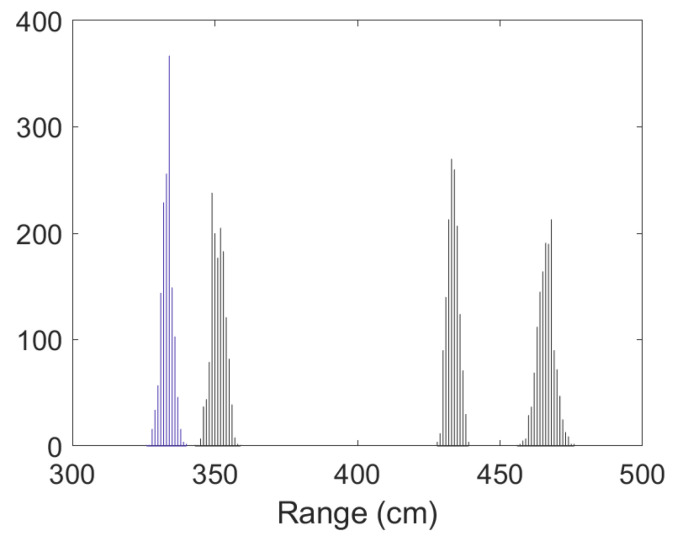
Scenario C2: Histogram of ranges.

**Figure 25 sensors-23-05660-f025:**
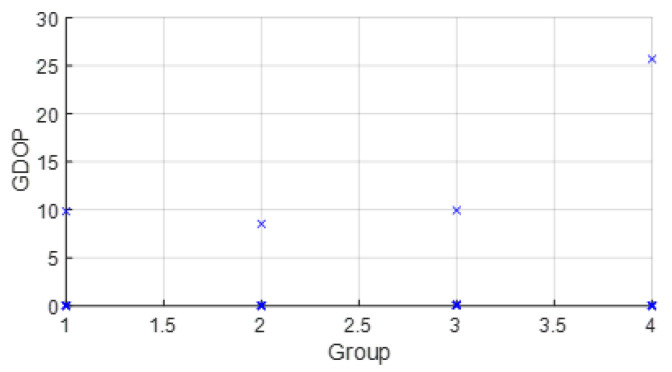
Scenario C2: WGDOP for each group of distances.

**Figure 26 sensors-23-05660-f026:**
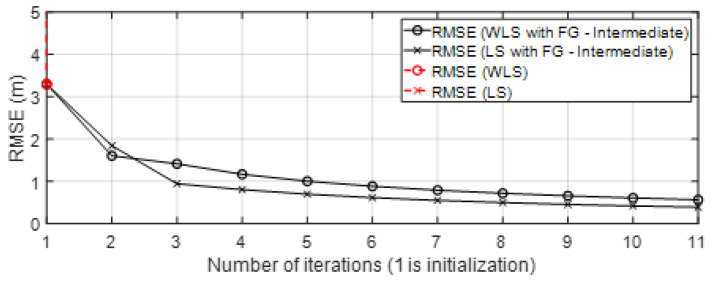
Scenario C2: RMSE.

**Table 1 sensors-23-05660-t001:** Main messages of the factor graph that avoids loops. This presents the operations of the main messages between components of the presented factor graph that avoids loops. The messages are shown in Figure 3.

Messages	N∼ (Random Variable; Mean, Covariance)
$\mu_{f_{d_{q}}\to \Delta d_{q}}$	$N∼(\Delta d_{q};{\underline{d}}_{d}-d_{d}^{0}$, $\sigma_{{\underline{d}}_{d}}^{2}),q=1,\ldots ,Q$
	d=1,…,D
$\mu_{\Delta d_{q}\to f_{\Delta d_{q}}}$	$\mu_{f_{d_{q}}\to \Delta d_{q}},q=1,\ldots ,Q$
$\mu_{f_{\Delta d_{q}}\to \Delta x_{q}}$	LS: $N∼(\Delta x_{q};{\left(H^{T}H\right)}^{-1}H^{T}m_{\Delta d_{q}},{\left(H^{T}H\right)}^{-1})$
	WLS: $N∼(\Delta x_{q};{\left(H^{T}WH\right)}^{-1}H^{T}Wm_{\Delta d_{q}},$
	${\left(H^{T}WH\right)}^{-1}),q=1,\ldots ,Q$
$\mu_{\Delta x\to f_{\Delta d_{x}}}$	$\prod_{q=1}^{Q}\mu_{f_{\Delta d_{q}}\to \Delta x}$
$\mu_{f_{\Delta x}\to x}$	$N∼(x;m_{\Delta x}+x^{0},\sigma_{\Delta x}^{2})$
$\mu_{x\to f_{x}}$	$\mu_{f_{\Delta x}\to x}$
$\mu_{f_{x}}\to x$	$N∼(x;x_{\mathrm{init}},\sigma_{x_{\mathrm{init}}}^{2})$

**Table 2 sensors-23-05660-t002:** Table of scenarios with the propagation conditions and geometry of the node placement.

Scenario	N (Number of Anchor Nodes)	Nodes	Conditions	Geometry
A1 (Challen.)	3	WALL1c;3;6	hard NLOS	Easy
A2 (Challen.)	4	WALL1d;7	hard NLOS	Medium
		1c;1b	LOS	
A3 (Challen.)	3	WALL1c;7;1b	hard NLOS	Medium
A4 (Challen.)	4	WALL1d;1b;1a	hard NLOS	Challen.
		1c	LOS	
B (Good)	4	WALL1c;3;5;6	LOS	Easy
C1 (Interm.)	4	1c;1d;1b;7	soft NLOS	Medium
C2 (Interm.)	4	1c;1d;1b;1a	soft NLOS	Challen.

**Table 3 sensors-23-05660-t003:** Table of scenario settings related to the distance error.

Scenario	N	Nodes	$\sigma_{{\underline{d}}_{i}}$ (m)
A1	3	WALL1c;3;6	(4.6369; 2.8964; 2.9455)/100
A2	4	WALL1c;1d;1b;7	(2.4827;4.7230;2.2134; 4.1242)/100
A3	3	WALL1c;1b;7	(2.5513;2.2483; 6.2294)/100
A4	4	WALL1c;1d;1b;1a	(1.8314;2.0573;2.1580; 2.7375)/100
B	4	WALL1c;3;5;6	(2.1805;2.0980;3.0475; 2.8364)/100
C1	4	WALL1c;1d;1b;7	(2.3584;2.8442;2.3765; 3.2908)/100
C2	4	WALL1c;1d;1b;1a	(1.9697;2.9474;2.4565; 2.0166)/100

## Data Availability

The datasets generated and analyzed during the current study are available in the Zenodo repository (see [23]).

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
