# Peer review of "A Data-Driven Factor Graph Model for Anchor-Based Positioning"

_sensors, 2023, doi:10.3390/s23125660_

Round 1

Reviewer 1 Report

I am willing to accept the manuscript in its current form, but I encourage the authors to reference and cite recent articles to improve the coherence of their arguments and augment their reference lists.

Reviewer 2 Report

This is an interesting paper, but you may improve this article to publish in this journal. Otherwise, I have a lot of recommendations to increase the quality of your paper. Be careful with the writing and mistakes.

Lines 5-6. When you write an acronym all the letters used to build it must be in capitals. So, you must write “Weighted Geometric Dilution Of Precision (WGDOP)” with the “O” of the word “of” in capitals as well.

Line 9. When you write an acronym, you must put in brackets its meaning. You must look for them in the whole manuscript and fix this tiny mistake. This is basically to make the reading easier for potential readers.

Line 15. At least, there is a keyword repeated in the article title. The keyword is “Factor Graph”. In order to increase the visibility of your paper I recommend changing this keyword. If you change it by another keyword, you will increase the probability that your paper could be found by future readers when they look for your paper in some databases like Scopus for example. If you repeat the same words in the article title and in keywords, less people could find your work. So, you must think about the visibility of your research.

Line 15. You must write the keywords in alphabetical order.

Line 27. When you write the references in the text and they are contiguous they must be in the same bracket, so, you must write as follows: “[3,4]”. Review this very common mistake your whole manuscript.

Lines 36-37. When you write an acronym all the letters used to build it must be in capitals. So, you must write “Non-Line Of Sight (NLOS)”. So, you must write the letter “O” of the word “of” in capitals as well.

Line 41. When you write the references in the text and they are contiguous they must be in the same bracket, so, you must write as follows: “[3,5]”. Just follow the rules of the journal. Review this very common mistake your whole manuscript.

Line 42. This sentence has no sense. Sometimes, when you write a reference you must write the authors as well in order to read the sentence with its meaning. So, you must write the authors close to the reference as well as follows: “As an example, Wymeersch et al. [6] applied Machine Learning (ML)…”. In this way the sentence is much more understandable and has more sense. Review this mistake in the whole paper.

Line 46. This sentence has no sense. Sometimes, when you write a reference you must write the authors as well in order to read the sentence with its meaning. So, you must write the authors close to the reference as well as follows: “In Dellaert and Kaess [8], a Factor Graph optimization…”. In this way the sentence is much more understandable and has more sense. Review this mistake in the whole paper.

Line 47. You must define an acronym “FG” just the very first time that you use it, so, you must define here, not after this line. Fix this mistake in your whole manuscript.

Line 47. This sentence has no sense. Sometimes, when you write a reference you must write the authors as well in order to read the sentence with its meaning. So, you must write the authors close to the reference as well as follows: “In Wen et al. [9], a FG was proposed…”. In this way the sentence is much more understandable and has more sense. Review this mistake in the whole paper.

Line 48. This sentence has no sense. Sometimes, when you write a reference you must write the authors as well in order to read the sentence with its meaning. So, you must write the authors close to the reference as well as follows: “…an FG based modelling: Bishop [10] explained how inference algorithms…”. Bishop is only an author himself, not several. In this way the sentence is much more understandable and has more sense. Review this mistake in the whole paper.

Line 51. This sentence has no sense. Sometimes, when you write a reference you must write the authors as well in order to read the sentence with its meaning. So, you must write the authors close to the reference as well as follows: “And Loeliger [11] explained FG based techniques for modelling…”. In this way the sentence is much more understandable and has more sense. Review this mistake in the whole paper.

Line 103. When you write an acronym, you must put in brackets its meaning. You must look for them in the whole manuscript and fix this tiny mistake. This is basically to make the reading easier for potential readers.

Line 135. I cannot see the “Section 1”.

You must write explicitly the section “Methods”.

Line 195. This sentence has no sense. Sometimes, when you write a reference you must write the authors as well in order to read the sentence with its meaning. So, you must write the authors close to the reference as well as follows: “In Moragrega [20], it is explained how the…”. Bishop is only an author himself, not several. In this way the sentence is much more understandable and has more sense. Review this mistake in the whole paper.

You must add the Discussion section as well comparing your research with other authors.

Otherwise, the authors adequately developed the Introduction, presenting the problems but you must write explicitly the objectives of this paper.

You must to write, as I told you before, the methods section explicitly.

The authors are to be congratulated for the results obtained in this article.

The quality of the English is high.

Reviewer 3 Report

Authors of this paper introduce a data-driven Factor Graph (FG) model designed to perform anchor-based indoor positioning. Particularly, this work is the continuation of authors’ previous work [17]. The content of the article meets with the topics of Sensors journal.

The basic topic of the article is interesting. However, the article has some parts that need extension or better explanation. The article needs more than major revision. Please, see my notes!

Notes:

o   Introduction – the state-of-the-art (SoA) is evaluated on good level. However, its first part, especially the Introduction, should be revised. There are mentioned several wireless systems used for indoor (and outdoor) localization purposes. There is mentioned IoT, but any one from this technology is not discussed or mentioned here. Maybe authors can provide higher attention to LoRa, which, among others, can be also employed for indoor and outdoor localization and tracking purposes. The authors should consider about the extension of SoA with appropriate work, for instance with “On the RSSI-based Indoor Localization employing LoRa in the 2.4 GHz ISM Band”, which provides performance study of LoRa-based indoor localization. Next, works dealing with applying of machine learning approaches to improve the accuracy of indoor localization should be also mentioned. For instance, I can recommend studying works like “RSS-Fingerprint Dimensionality Reduction for Multiple Service Set Identifier-Based Indoor Positioning Systems” and “Received Signal Strength Fingerprinting-Based Indoor Location Estimation Employing Machine Learning”. I hope that the mentioned work can be helpful in the improvement of the elaboration of SoA and can present helpful information for potential readers. Otherwise, please, work with other paper. Thank you!

o   Article – please, describe better that what was the main motivation to use the UWB technology

o   Section 2 – Figs. 1 and 2 are the same ones that were used in [17]. Hence, problems with IEEE copyright can occur.

o   Sections 2 and 3 – it is hard to find what the main or new contribution of this article is related to the theory background. This work is strongly based on [17] and there are many parts which are very similar. Moreover, the Algorithm 1 is very similar to the previously introduced solution.

o   Section 4 – it is not clear that on the base what the simulation parameters were selected

o   Section 4 – please, describe the measurement environment and the offline data processing in detail (for inspiration, you can check the work “Received Signal Strength Fingerprinting-Based Indoor Location Estimation Employing Machine Learning”). Next, please, give more information about the data collection. What about measurement setup?

o   Section 4 – for reproducible research, I would suggest the authors make the HW/SW source codes publicly available

o   Section 4 – the obtained results and main outputs of the article should be better compared with similar SoA studies (solutions).

o   Appendix – what is the difference between this Appendix and the Appendix presented in [17]? Is there any serious difference?

The English grammar of the article contains some minor typos – not critical. Hence, please, check the whole article carefully once again!

Round 2

Reviewer 3 Report

The article has been partly improved. Thanks for the explanation letter. However,next time, do not forget to highlight all changes in the revised article. Thanks!

Notes:

- the main outputs of the article should be better discussed in terms of state-of-the-art

- Appendix -- Please, consider about the removing of appendix and replace it with a link on your previous article -- you have mentioned that the same appendix is available in other article
